# Chlorine Radical-Initiated Atmospheric Oxidation of Imines: Implications for Structural Influence on the Nitrosamine Formation

Qian Xu<sup>1</sup>, Fangfang Ma<sup>1,2</sup>\*, Chang Liu<sup>1</sup>, Qiaojing Zhao<sup>1</sup>, Jingwen Chen<sup>1</sup>, Hong-Bin Xie<sup>1</sup>\*

<sup>1</sup>Key Laboratory of Industrial Ecology and Environmental Engineering (Ministry of Education), School of Environmental Science and Technology, Dalian University of Technology, Dalian 116024, China <sup>2</sup>College of Resources and Environmental Engineering, Guizhou Provincial Key Laboratory for Prevention and Control of Emerging Contaminants, Guizhou University, Guiyang, 550025, PR China

10 Correspondence to: ffma@gzu.edu.cn (Fangfang Ma) and hbxie@dlut.edu.cn (Hong-Bin Xie)

Abstract. Chlorine radical (·Cl) initiated oxidation of organic nitrogen compounds (ONCs) plays an important role in carcinogenic nitrosamines formation. Imines are important constituents of ONCs, primarily formed from the atmospheric oxidation of amines. However, ·Cl-initiated atmospheric oxidation of imines remains poorly understood. Here, we studied the reaction mechanisms and kinetics of ·Cl-initiated oxidation for five representative imines (CH<sub>2</sub>=NH, CH<sub>3</sub>CH=NH, CH<sub>3</sub>N=CH<sub>2</sub>, (CH<sub>3</sub>)<sub>2</sub>C=NH, HN=CHCH<sub>2</sub>OH) to elucidate their atmospheric fate and extend the limited available data of ONCs, thereby establishing a structure-activity relationship for the reactions. The calculated overall reaction rate constants (× 10<sup>-11</sup> cm³ molecule<sup>-1</sup> s<sup>-1</sup>) of ·Cl + CH₂=NH, ·Cl + CH₃CH=NH, ·Cl + CH<sub>3</sub>N=CH<sub>2</sub>, ·Cl + (CH<sub>3</sub>)<sub>2</sub>C=NH, and ·Cl + HN=CHCH<sub>2</sub>OH are 4.5 、27.2 、7.32 、44.8 and 12.6, respectively, which are consistent with the available experimental values. Importantly, our results show that the 'Cl-initiated reactions of the NH-containing imines mainly produce N-centered radicals. These N-centered radicals exhibit various fates under tropospheric conditions: mainly reacting with NO to form nitrosamines or with O<sub>2</sub> to form cyanide compounds, which differs substantially from the behavior of previously reported amines. The various fates of the N-centered radicals formed from imines originates from the difference in direct hydrogen abstraction reaction rate constants ( $k_{O2}$ ) with  $O_2$  and the reaction rate  $(k_{NO})$  with NO, both of which are principally governed by the distinct molecular structure of Ncentered radicals. The revealed reaction mechanism provides new insights into the atmospheric transformation and risks of imines, and enrich our understanding of ·Cl/ONCs chemistry.

### 1 Introduction

Chlorine radicals (·Cl) are key atmospheric oxidants that significantly impact air quality and radical budgets by oxidizing volatile organic compounds (VOCs), thereby contributing to the formation of O<sub>3</sub>, ·OH and secondary organic aerosol (SOA), and consequent impacts on climate (Gunthe et al., 2021; Liu et al., 2024; Wang et al., 2019; Yi et al., 2023; Cao et al., 2024). Historically, ·Cl was considered to originate mainly from heterogeneous reactions on sea salt particles (Faxon and Allen, 2013). In recent years, diverse sources of ·Cl precursors have been identified in urban and suburban environments, e.g.,

coal combustion, biomass burning, road salt application, chlorine-based fertilizers, and automotive braking processes (Li et al., 2020; Fu et al., 2018; Li et al., 2024; Yin et al., 2022; Thornton et al., 2010). Daytime ·Cl concentrations have been reported to reach up to 10<sup>6</sup> molecule cm<sup>-3</sup>, comparable to typical ·OH concentrations (Young et al., 2014). Laboratory and theoretical studies have shown that ·Cl-initiated oxidation of VOCs proceeds 10-100 times faster than those initiated by ·OH (Edwards and Young, 2024). Moreover, ·Cl-initiated oxidation of VOCs often follows distinct pathways and produce products that differ significantly from those of ·OH-initiated reactions (Chen et al., 2025; Guo et al., 2020; Wang et al., 2022). For example, ·Cl-initiated oxidation of VOCs could yield chlorine-containing products, potentially altering their toxicity (Fu et al., 2024; Ding et al., 2021). Thus, ·Cl-initiated reactions represent a crucial role in governing the fate of VOCs and warrant careful consideration.

Organic nitrogen compounds (ONCs), a major subclass of VOCs, account for approximately 65% of VOCs and are commonly detected in the atmosphere (Abudumutailifu et al., 2024). ONCs play significant roles in the formation of particles and hazardous substances, such as hydrogen cyanide (HCN), nitrous oxide (N<sub>2</sub>O), nitrosamines, and nitramines (Ning et al., 2022; Wang et al., 2023; Liu et al., 2021; Abudumutailifu et al., 2024; Sun et al., 2024; Elm et al., 2016). Imines (R<sub>1</sub>R<sub>2</sub>C=NR<sub>3</sub>) comprise about 16% of atmospheric ONCs (Ditto et al., 2022), which originate from multiple sources, including combustion, motor vehicles, cleaning sports equipment and the atmospheric oxidation of amines, the latter being the primary one(Waterman and Hillhouse, 2008; Zhu et al., 2022; You et al., 2022; Onel et al., 2014). In urban areas, imines have been detected at ppt levels, comparable to typical concentrations of amines (Zhu et al., 2022). Given their abundance and their role as first-generation oxidation products of amines, elucidating the atmospheric reaction mechanisms of imines is essential for accurately assessing their environmental impacts.

To date, only limited studies have investigated the atmospheric chemistry of imines (Ali, 2020; Xu et al., 2024; Ali et al., 2016; Bunkan et al., 2014; Akbar Ali et al., 2018; Bunkan et al., 2022; Almeida and Kurtén, 2022; Ditto et al., 2020; Yao et al., 2016). Atmospheric oxidation by ·OH and ·Cl are important removal pathways for imines. Previous experimental and theoretical studies have primarily examined the reaction mechanism and kinetics of ·OH-initiated reactions with simple imines, such as methylimine (CH<sub>2</sub>NH) (Bunkan et al., 2014; Akbar Ali et al., 2018), N-methylmethylimine (CH<sub>3</sub>NCH<sub>2</sub>) (Bunkan et al., 2022), and cyclic imine 1,2,3,6-tetrahydropyrazine (THPyz) (Almeida and Kurtén, 2022). These studies demonstrated that the substituents on the C=N bond profoundly influence the reactivity of imines, ultimately leading to varying atmospheric impacts. Nevertheless, no previous studies have addressed ·Cl-initiated reactions of imines. Since ·OH and ·Cl-initiated reactions of ONCs follow distinct mechanisms and yielding different products (Xie et al., 2015; Xue et al., 2022; Xie et al., 2014), the reactivity of imines toward ·Cl cannot be directly inferred from their ·OH chemistry.

Our previous studies demonstrated that  $\cdot$ Cl exhibits a particularly strong interaction with the NH<sub>x</sub> group of ONCs, leading to the formation of N-centered radicals, which are recognized precursors of carcinogenic nitrosamines (Xie et al., 2017; Liu et al., 2019). Additionally, the yield of N-centered radicals is strongly dependent on the structure of ONCs. Given the electronic structures of imines,  $\cdot$ Clinitiated atmospheric oxidation of imines has the potential to form N-centered radicals. However, the nitrogen lone pair electrons of imines can be delocalized to some extent to their adjacent C=N bond,

which tends to decrease its electron-donor ability compared with previously well-studied ONCs (Carey and Sundberg, 2007), which may alter the reaction mechanisms of ·Cl + imines reactions. Moreover, the subsequent reactions of N-centered radicals depended on their specific structures. Therefore, to broaden our comprehension of the atmospheric chemistry of imines and fully assess their environmental risks, it is essential to elucidate the reaction mechanism and kinetics of ·Cl + imines reactions.

In this study, the reaction mechanisms and kinetics of imines with  $\cdot$ Cl were investigated by selecting five simple imines {i.e., methanimine (CH<sub>2</sub>=NH), ethanimine (CH<sub>3</sub>CH=NH), N-methylmethanimine (CH<sub>3</sub>N=CH<sub>2</sub>), 2-propanimine ((CH<sub>3</sub>)<sub>2</sub>C=NH), and 2-iminoethanol (HN=CHCH<sub>2</sub>OH)} as model compounds. A combination of quantum chemical calculations and kinetic modeling was employed to elucidate these reactions. The parent amines of all selected imines have been detected in the ambient atmosphere (Qiu and Zhang, 2013; Ge et al., 2011; Shen et al., 2023). For the imine radicals yielded in the initial reaction step, their subsequent reactions including isomerization/dissociation, and bimolecular reactions with key atmospheric oxidants (O<sub>2</sub> and NO) were investigated. These results provide valuable insights into the reaction mechanisms and kinetics of imines, thereby advancing our understanding of their atmospheric chemistry and  $\cdot$ Cl chemistry.

### 2 Computational Details

**2.1 Global Minimum Search.** The selected imines can exist in multiple distinct gas-phase configurations. To identify their global minima, a multi-step conformational sampling scheme was employed, following our earlier studies (Lu et al., 2024; Ma et al., 2019; Ma et al., 2023) To explore their conformational space, the gentor module within the Molclus program was initially used to form the range of conformations. The generated conformers were further optimized at the PM6 and MP2/6-31+G(3df,2p) level (Vereecken and Francisco, 2012). Single-point energy calculations were performed at the CCSD(T)/aug-cc-pVTZ level (Vereecken and Francisco, 2012). The structure with the lowest Gibbs free energy were determined to be the global minima and served as the initial structure for investigating the reaction mechanisms and kinetics. The corresponding global minima of five imines are presented in Figure S1.

**2.2 Electronic Structure Calculations.** All electronic structure calculations were conducted with the Gaussian 09 program package (Frisch et al., 2009). Consistent with our previous studies, the initial reactions of 'Cl with imines' were investigated by performing geometry optimizations and harmonic vibrational frequency calculations at the MP2/6-31+G(3df,2p) level, followed by single-point energy evaluations at the CCSD(T)/aug-cc-pVTZ level. For the subsequent reactions of the imine radicals formed in the initial step, geometric structures and vibrational frequencies calculations were carried out at the M06-2X/6-31+G(d,p) level (Zhao and Truhlar, 2008), and single-point energy calculations were performed at the CCSD(T)/6-311+G(2df,2p) level (Pople et al., 1989). Intrinsic reaction coordinate (IRC) calculations were used to confirm that each transition state correctly connects the relevant reactants and products at the respective optimization levels (Fukui, 1981). The isolated 'Cl was treated with a correction value of 0.8 kcal mol<sup>-1</sup> to account for spin-orbit coupling effects (Ma et al., 2018). Natural bond orbital (NBO) analysis was conducted to characterize the atomic charges in the initial reaction transition states

125

145

150

(Reed et al., 1985). Unless otherwise specified, the following labels are used throughout the manuscript: 'reactants' (R), 'transition states' (TS), 'intermediates' (IM) and 'products' (P).

2.3 Kinetics Calculations. The reaction kinetics for the initial reaction and subsequent reactions were calculated using the MultiWell 2014.1 and MESMER 6.0 software packages, respectively (Barker et al., 2014; Barker, 2001a; Glowacki et al., 2012). Rate constants for the multi-channel and multi-well chemical reactions involving tight transition states were calculated using Rice-Ramsberger-Kassel-Marcus (RRKM) theory (Barker, 2001b), while those for barrierless reactions were obtained with long-range transition-state theory with a dispersion force potential or Inverse Laplace Transformation (ILT) model. Full computational settings are provided in our previous studies (Ma et al., 2018; Ma et al., 2021). For single-step processes, canonical transition state theory (TST) in the Thermo module of MultiWell-2014.1 program suite was applied to calculate the rate constants. Tunneling corrections for H-shift or H-abstraction reactions were included via a one-dimensional asymmetric Eckart potential (Eckart, 1930). Reaction rate constants for ·Cl + imines reactions were performed across the temperature range of 260 – 330 K. Lennard-Jones parameters for various species used in the MultiWell or MESMER are shown in Table S1.

### 3 Results and Discussion

**3.1 ·Cl-Initiated Reactions.** ·Cl can either abstract H atoms from the = $CH_{x^-}$  ( $x = 1 \sim 2$ ) and =NH groups or add to the C=N bond of the target imines. The calculated energetic data are summarized in Table 1, with the corresponding zero point energy (ZPE) corrected potential energy surfaces (PES) shown in Figure S2. Based on the reaction activation energies (Ea) in Table 1, it can be concluded that Habstractions at the N-sites is the most energetically favorable pathway for Cl-initiated reactions of 135 CH<sub>2</sub>=NH, CH<sub>3</sub>CH=NH, (CH<sub>3</sub>)<sub>2</sub>C=NH, and HN=CHCH<sub>2</sub>OH, which is similar to amines + ·Cl systems (Ma et al., 2018; Xue et al., 2022; Xie et al., 2015; Xie et al., 2017). Notably, the  $E_a$  values for the formation of N-centered radicals from imines are significantly higher than those from amines. Interestingly, OH-initiated reactions of CH<sub>2</sub>=NH can also result in the formation of N-centered radicals, unlike ·OH-initiated reactions of amines, although the  $E_a$  value for the ·OH + CH<sub>2</sub>NH reaction being 140 approximately 4 kcal mol<sup>-1</sup> higher than that of ·Cl + CH<sub>2</sub>=NH reaction (Bunkan et al., 2014). For CH<sub>3</sub>N=CH<sub>2</sub>, H-abstraction from the -CH<sub>2</sub> site forming C-centered radicals is the most favorable pathway, consistent with ·OH + CH<sub>3</sub>N=CH<sub>2</sub> system (Bunkan et al., 2022).

It should be noted that, despite numerous attempts, the TSs for H-abstraction from the  $CH_2$  site of  $CH_3N=CH_2$  and for  $\cdot Cl$  addition to the N site of  $(CH_3)_2C=NH$  could not be located at the MP2/6-31+G(3df,2p) level. By analyzing the thermodynamic data, we found that  $\cdot Cl$  addition to the N-site is thermodynamically unfavorable for the  $\cdot Cl + (CH_3)_2C=NH$  reactions, suggesting that this pathway plays a negligible role. For  $(CH_3)_2C=NH$ , the TS  $(TS_{3-2})$  of  $\cdot Cl$  abstracting an H-atom from  $CH_2$  site was located at the MP2/6-31+G(d,p) level. To evaluate the performance of the MP2/6-31+G(d,p) method for geometry optimization and CCSD(T)/6-311+G(2df,2p) for single point energy calculation in calculating the reaction energies  $(E_a$  and  $\Delta E)$  of H-abstraction at the  $CH_2$  site of  $CH_3N=CH_2$ , we randomly selected

160

two pathways to calculate the  $E_a$  and  $\Delta E$  values using the CCSD(T)/6-311+G(2df,2p)//MP2/6-31+G(d,p) and CCSD(T)/aug-cc-pVTZ//MP2/6-31+G(3df,2p) methods. The results show that the difference in  $E_a$  and  $\Delta E$  values between the CCSD(T)/aug-cc-pVTZ//MP2/6-31+G(3df,2p) and CCSD(T)/6-311+g(2df,2p)//MP2/6-31+G(d,p) were within the quantum chemistry method (1.0 mol<sup>-1</sup>) (Table S2). These indicate that the more efficient CCSD(T)/6-311+g(2df,2p)//MP2/6-31+G(d,p) method is reliable, and the  $E_a$  values of TS<sub>3-2</sub> obtained with this method have negligible effects on the conclusions.

Table 1. Calculated reaction activation energies ( $E_a$ , in kcal mol<sup>-1</sup>), thermodynamic energies ( $\Delta E$ , kcal mol<sup>-1</sup>), branch ratios  $\Gamma$  and reaction rate constants ( $k_{Cl}$ , in cm<sup>3</sup> molecule<sup>-1</sup> s<sup>-1</sup>) for the reactions of ·Cl with (A) CH<sub>2</sub>=NH, (B) CH<sub>3</sub>CH=NH, (C) CH<sub>3</sub>N=CH<sub>2</sub>, (D) (CH<sub>3</sub>)<sub>2</sub>C=NH, and (E) HN=CHCH<sub>2</sub>OH at the CCSD(T)/aug-cc-pVTZ//MP2/6-31+G(3df,2p) level of theory.

| Species            | TS                    | $E_{ m a}$ | $\Delta E$ | Γ           | $k_{\rm Cl} \times 10^{-10}$ |
|--------------------|-----------------------|------------|------------|-------------|------------------------------|
| P <sub>1-1</sub>   | $TS_{1-1}$            | 2.35       | 0.08       | 0.26%       |                              |
| $P_{1-2}$          | $TS_{1-2}$            | -0.60      | -4.30      | 8.25%       |                              |
| $P_{1-3}$          | $TS_{1-3}$            | -4.39      | -12.22     | 90.85%      | 0.45                         |
| $P_{1-4}$          | $TS_{1-4}$            | 0.89       | -13.13     | 0.64%       |                              |
| $P_{1-5}$          | $TS_{1-5}$            | 5.14       | 8.97       | 0.00%       |                              |
| P <sub>2-1</sub>   | TS <sub>2-1</sub>     | 7.95       | 5.68       | 0.00%       |                              |
| $P_{2-2/3}$        | $TS_{2-2}$            | 2.26       | -8.59      | 0.06%       |                              |
| $P_{2-2/3}$        | $TS_{2-3}$            | 2.29       | -8.59      | 0.05%       |                              |
| P <sub>2-4</sub>   | $TS_{2-4}$            | -2.22      | -5.15      | 11.96%      | 2.72                         |
| P <sub>2-5</sub>   | $TS_{2-5}$            | -5.99      | -10.55     | 87.59%      |                              |
| P <sub>2-6</sub>   | $TS_{2-6}$            | -0.85      | -11.12     | 0.35%       |                              |
| P <sub>2-7</sub>   | TS <sub>2-7</sub>     | 6.57       | 8.46       | 0.00%       |                              |
| $P_{3-1}$          | $TS_{3-1}$            | 1.91       | 1.87       | 0.16%       |                              |
| P <sub>3-2</sub>   | $TS_{3\text{-}2}{}^a$ | -2.57      | -1.66      | 59.36%      |                              |
| $P_{3-3}$          | $TS_{3-3}$            | -1.67      | -2.01      | 32.98%      | 0.73                         |
| $P_{3-4/5}$        | $TS_{3-4/5}$          | 1.39       | -12.00     | 0.98%/0.98% | 0.73                         |
| P <sub>3-6</sub>   | $TS_{3-6}$            | -1.22      | -15.02     | 5.53%       |                              |
| P <sub>3-7</sub>   | $TS_{3-7}$            | 4.26       | 8.19       | 0.00%       |                              |
| $P_{4-1}$          | $TS_{4-1}$            | 6.31       | 4.35       | 0.00%       |                              |
| $P_{4-2/3}$        | $TS_{4-2/3}$          | 0.89       | -6.82      | 0.01%/0.01% |                              |
| $P_{4-4}$          | $TS_{4-4}$            | 4.19       | 2.94       | 0.00%       |                              |
| $P_{4-5/6}$        | $TS_{4\text{-}5/6}$   | 1.21       | -7.13      | 0.01%/0.01% | 4.48                         |
| P <sub>4-7</sub>   | $TS_{4-7}$            | -6.87      | -10.35     | 99.97%      |                              |
| $P_{4-8}$          | $TS_{4-8}$            | -1.40      | -10.53     | 0.00%       |                              |
| P <sub>4-9</sub>   | TS <sub>4-9</sub>     |            | 12.43      |             |                              |
| P <sub>5-1/2</sub> | TS <sub>5-1/2</sub>   | -1.13      | -20.52     | 6.08%/6.08% |                              |
| P <sub>5-3</sub>   | $TS_{5-3}$            | -1.53      | -2.98      | 39.46%      | 1.26                         |
| P <sub>5-4</sub>   | $TS_{5-4}$            | -2.59      | -6.96      | 48.33%      |                              |
|                    |                       |            |            |             |                              |

| P <sub>5-6</sub> TS <sub>5-6</sub> 1.16 -8.83 0.05%<br>P <sub>5-7</sub> TS <sub>5-7</sub> 9.42 12.23 0.00% |
|------------------------------------------------------------------------------------------------------------|
|                                                                                                            |
| 15-5 155-5 15.07 7.07 0.0070                                                                               |
| P <sub>5-5</sub> TS <sub>5-5</sub> 15.09 9.69 0.00%                                                        |

<sup>a</sup>TS<sub>3-2</sub> was computed at the CCSD(T)/6-311+G(2df,2p)//MP2/6-31+G(d,p) level.

Since ·Cl abstract H-atom from sp<sup>2</sup>-N site is more favorable than from other sites, it merits further discussion why the  $E_a$  values for generating N-centered radicals are substantially lower than those for C-centered radicals. By analyzing NBO charges for all TSs, we found that larger charge transfers occurred at the most favorable transition states  $TS_{1-3}$  (-0.428 e),  $TS_{2-5}$  (-0.447 e),  $TS_{3-2}$  (-0.318 e) and  $TS_{4-7}$  (-0.456 e) than other TSs (see Table S3). This indicates that charge transfer contributes critically to the stabilization of these transition states, thereby facilitating the formation of N-centered radicals in ·Clinitiated reactions of NH-containing imines. A similar phenomenon was observed in the piperazine (PZ) + ·Cl system (Ma et al., 2018). However, in the case of ·Cl + HN=CHCH<sub>2</sub>OH system, the charge transfer at the  $TS_{5-5}$  (-0.430 e) is slightly larger than that at the  $TS_{5-4}$  (the most favorable one, -0.423 e) (see Table S3). The presence of intermolecular hydrogen bonds in  $TS_{5-4}$  may account for its lower  $E_a$  value (see Figure S2).

Using the master equation approach, the overall rate constants ( $k_{\rm Cl}$ ) were determined to be 4.50 ×  $10^{-11}$ ,  $2.72 \times 10^{-10}$ ,  $7.32 \times 10^{-11}$ ,  $4.48 \times 10^{-10}$  and  $1.26 \times 10^{-10}$  cm<sup>3</sup> molecule<sup>-1</sup> s<sup>-1</sup> for ·Cl + CH<sub>2</sub>=NH, ·Cl + CH<sub>3</sub>CH=NH, ·Cl + CH<sub>3</sub>N=CH<sub>2</sub>, ·Cl + (CH<sub>3</sub>)<sub>2</sub>C=NH, and ·Cl + HN=CHCH<sub>2</sub>OH reactions at 298 K and 1 atm, respectively. The experimental  $k_{\rm Cl}$  value available for the CH<sub>3</sub>N=CH<sub>2</sub> + ·Cl reaction is (1.9 ± 0.15) ×  $10^{-11}$  cm<sup>3</sup> molecule<sup>-1</sup> s<sup>-1</sup>(Bunkan et al., 2022), which in good consistency with the corresponding computational result (7.32 ×  $10^{-11}$  cm<sup>3</sup> molecule<sup>-1</sup> s<sup>-1</sup>). This could further support the reliability of our computational approach. Over the temperature range of 260-330 K, CH<sub>2</sub>=NH and (CH<sub>3</sub>)<sub>2</sub>C=NH show positive temperature dependence for  $k_{\rm Cl}$  (Figure S3A and S3D), whereas CH<sub>3</sub>CH=NH, CH<sub>3</sub>N=CH<sub>2</sub> and HN=CHCH<sub>2</sub>OH exhibit a negative dependence (Figure S3B, S3C and S3E). By analyzing the substitutions effects on the  $k_{\rm Cl}$  values, it can be found that CH<sub>3</sub> and (CH<sub>3</sub>)<sub>2</sub> substitutions at the >CR1= position, as well as the CH<sub>3</sub> substitution at the =NR2 position, increase the  $k_{\rm Cl}$  values, while substitutions at the R3CCH= position have little effect on the  $k_{\rm Cl}$  values.

The calculated branch ratios ( $\Gamma$  values) for the N-centered radicals in the reactions of  $\cdot$ Cl + CH<sub>2</sub>=NH,  $\cdot$ Cl + CH<sub>3</sub>CH=NH,  $\cdot$ Cl + (CH<sub>3</sub>)<sub>2</sub>C=NH and  $\cdot$ Cl + HN=CHCH<sub>2</sub>OH are 90.85%, 87.59%, 99.97% and 48.33% at 1 atm and 298 K, respectively. The  $\Gamma$  values of N-centered radicals show a slight decrease with increasing temperature, whereas those of other product species remain very small and negligible across the studied temperature range (Figure S4A, S4B, S4D and S4E). Therefore, N-centered radicals are the main products in these four reactions under atmospheric conditions, similar to  $\cdot$ Cl + amines systems (Ma et al., 2018; Xie et al., 2015). For the  $\cdot$ Cl + CH<sub>3</sub>N=CH<sub>2</sub> reaction, the calculated  $\Gamma$  values for P<sub>3-1</sub>, P<sub>3-2</sub>, P<sub>3-3</sub>, P<sub>3-4/5</sub>, P<sub>3-6</sub>, and P<sub>3-7</sub> are 0.16%, 59.36%, 32.98%, 0.98%, 5.53%, 0.00%, respectively, indicating a strong preference for the formation of C-centered radicals (Figure S4C). Since previous study have investigated the transformation of CH<sub>3</sub>N=CH· (P<sub>3-2</sub>) (Bunkan et al., 2022), we mainly considered the further transformation of the four N-centered radicals formed in the subsequent sections.

https://doi.org/10.5194/egusphere-2025-4896 Preprint. Discussion started: 22 October 2025 © Author(s) 2025. CC BY 4.0 License.

## 3.2 Subsequent Reactions of the formed N-centered radicals.

Consistent with previously studied N-centered radicals (Ma et al., 2018; Xie et al., 2015; Da Silva, 2013; Tang and Nielsen, 2012), the four N-centered radicals (CH<sub>2</sub>=N·, CH<sub>3</sub>CH=N·, (CH<sub>3</sub>)<sub>2</sub>C=N·, and ·N=CHCH<sub>2</sub>OH) will subsequently undergo self-isomerization/dissociation or react with key atmospheric oxidants, such as O<sub>2</sub> and NO. From the calculated ZPE-corrected PES of self-isomerization/dissociation for these four N-centered radicals (Figure S5), it can be found that the  $E_a$  values are in the range of 27.85 ~ 79.58 kcal mol<sup>-1</sup>, suggesting that both the self-isomerization and dissociation processes proceed very slowly. Therefore, in the atmosphere, these four N-centered radicals are more likely to react with O<sub>2</sub> and NO.

Regarding the reactions of the four N-centered radicals with  $O_2$ , two distinct routes were identified. The first pathway involves  $O_2$  directly abstracting an H atom from = $CH_{x-}$  ( $x = 1 \sim 2$ ) and  $-CH_3$  sites, forming cyanide compounds, cyclic imines and  $HO_2$ . The second pathway proceeds by  $O_2$  addition to the =N- site, with  $E_a$  values in the range of 5.60  $\sim$  14.18 kcal mol<sup>-1</sup>, forming adducts with various conformations depending on the direction of  $O_2$  attack. The calculated ZPE-corrected PES for the four N-centered radicals +  $O_2$  reactions are shown in Figure 1. With the exception of  $(CH_3)_2C=N$ , the  $E_a$  values for the direct H-abstraction pathways in the  $CH_2=N$  +  $O_2$ ,  $CH_3CH=N$  +  $O_2$ , and  $N=CHCH_2OH$  +  $O_2$  reactions are 12.25, 10.86 and 10.89 kcal mol<sup>-1</sup>, respectively, significantly lower than those of addition pathways. Therefore, the H-abstraction pathway represents the primary pathway for these three reactions, producing cyanide compounds and  $HO_2$ . A similar direct H-abstraction mechanism has been observed in the reactions of N-centered radicals formed from amines with  $O_2$  (Ma et al., 2018; Xue et al., 2022; Xie et al., 2015; Xie et al., 2017).

For  $(CH_3)_2C=N$ , the  $E_a$  value of  $O_2$  addition forming  $IM_{12-1}$  is lower than that for the direct H-abstraction pathways. Therefore, the formation of adduct  $IM_{12-1}$  is most favorable in the initial attack of  $O_2$ . The subsequent unimolecular H-shift reaction of  $IM_{12-1}$  requires overcoming a high barrier of 31.89 kcal mol<sup>-1</sup>, corresponding to a rate constant of  $9.35 \times 10^{-10}$  s<sup>-1</sup> at 298 K. This intramolecular process is slower than the pseudo-first-order rates for the bimolecular reactions of  $IM_{12-1}$  with NO and  $HO_2$ , which are estimated as 1.25 s<sup>-1</sup> and 0.02 s<sup>-1</sup>, respectively, under typical atmospheric conditions (5 ppb NO and 50 ppt  $HO_2$ ), based on bimolecular rate constants of  $k_{NO} = 1.0 \times 10^{-11}$  cm<sup>3</sup> molecule<sup>-1</sup> s<sup>-1</sup> (Atkinson and Arey, 2003; Atkinson, 2000). Therefore,  $IM_{12-1}$  will primarily undergo bimolecular reactions with NO and  $HO_2$ , yielding alkoxy radical, organic nitrites and hydroperoxides as the main products.

Figure 1. Schematic ZPE-corrected PES for the reactions of  $O_2$  with (A)  $CH_2=N^{\bullet}$ , (B)  $CH_3CH=N^{\bullet}$ , (C)  $(CH_3)_2C=N^{\bullet}$ , and (D)  $N=CHCH_2OH$  at the CCSD(T)/6-311+G(2df,2p)//M06-2X/6-31+G(d,p) level of theory.

Our previous studies revealed that Ea values for O2 directly abstracting H-atom of N-centered radicals are highly sensitive to the employed theorical methods, which ultimately affects the reaction rate constant ( $k_{O2}$ ) for the reactions with O<sub>2</sub> (Liu et al., 2019; Xie et al., 2017). Since  $k_{O2}$  is a key parameter in determining nitrosamines yields in subsequent reactions, we further evaluated the impact of different computational approaches on the  $E_a$  values for direct H-abstraction pathways (Liu et al., 2019; Wang, 2015). The E<sub>a</sub> values obtained at CCSD(T)/aug-cc-pVTZ/MP2/6-31+G(3df,2p), CCSD(T)/aug-cc-pVTZ/MP2/AUg-cc-pVTZ/MP2/AUg-cc-pVTZ/MP2/AUg-cc-pVTZ/MP2/AUg-cc-pVTZ/MP2/AUg-cc-pVTZ/MP2/AUg-cc-pVTZ/MP2/AUg-cc-pVTZ/MP2/AUg-cc-pVTZ/MP2/AUg-cc-pVTZ/MP2/AUg-cc-pVTZ/MP2/AUg-cc-pVTZ/MP2/AUg-cc-pVTZ pVTZ//MP2/aug-cc-pVTZ and CCSD(T)/6-311+G(2df,2p)//M06-2X/6-31+G(d,p) are summarized in Table S4. When the geometry optimization method was changed from M06-2X/6-31+G(d,p) to MP2/6-31+G(3df,2p) or MP2/aug-cc-pVTZ, and the single point energy calculation was changed from CCSD(T)/6-311+G(2df,2p) to CCSD(T)/aug-cc-pVTZ, the deviations in  $E_a$  values reach up to 6.2 kcal  $mol^{-1}$  [between CCSD(T)/aug-cc-pVTZ//MP2/6-31+G(3df,2p) and CCSD(T)/6-311+G(2df,2p)//M06-31+G(2df,2p) 2X/6-31+G(d,p)] and 6.4 kcal  $mol^{-1}$  [between CCSD(T)/aug-cc-pVTZ//MP2/aug-cc-pVTZ and CCSD(T)/6-311+G(2df,2p)//M06-2X/6-31+G(d,p)]. By contrast, changing the optimization method from MP2/6-31+G(3df,2p) to MP2/aug-cc-pVTZ lead to only 0.3 kcal mol<sup>-1</sup> deviation of  $E_a$  values  $between \quad CCSD(T)/aug-cc-pVTZ//MP2/6-31+G(3df,2p) \quad and \quad CCSD(T)/aug-cc-pVTZ//MP2/aug-cc-pVTZ//MP2/aug-cc-pVTZ//MP2/aug-cc-pVTZ//MP2/aug-cc-pVTZ//MP2/aug-cc-pVTZ//MP2/aug-cc-pVTZ//MP2/aug-cc-pVTZ//MP2/aug-cc-pVTZ//MP2/aug-cc-pVTZ//MP2/aug-cc-pVTZ//MP2/aug-cc-pVTZ//MP2/aug-cc-pVTZ//MP2/aug-cc-pVTZ//MP2/aug-cc-pVTZ//MP2/aug-cc-pVTZ//MP2/aug-cc-pVTZ//MP2/aug-cc-pVTZ//MP2/aug-cc-pVTZ//MP2/aug-cc-pVTZ//MP2/aug-cc-pVTZ//MP2/aug-cc-pVTZ//MP2/aug-cc-pVTZ//MP2/aug-cc-pVTZ//MP2/aug-cc-pVTZ//MP2/aug-cc-pVTZ//MP2/aug-cc-pVTZ//MP2/aug-cc-pVTZ//MP2/aug-cc-pVTZ//MP2/aug-cc-pVTZ//MP2/aug-cc-pVTZ//MP2/aug-cc-pVTZ//MP2/aug-cc-pVTZ//MP2/aug-cc-pVTZ//MP2/aug-cc-pVTZ//MP2/aug-cc-pVTZ//MP2/aug-cc-pVTZ//MP2/aug-cc-pVTZ//MP2/aug-cc-pVTZ//MP2/aug-cc-pVTZ//MP2/aug-cc-pVTZ//MP2/aug-cc-pVTZ//MP2/aug-cc-pVTZ//MP2/aug-cc-pVTZ//MP2/aug-cc-pVTZ//MP2/aug-cc-pVTZ//MP2/aug-cc-pVTZ//MP2/aug-cc-pVTZ//MP2/aug-cc-pVTZ//MP2/aug-cc-pVTZ//MP2/aug-cc-pVTZ//MP2/aug-cc-pVTZ//MP2/aug-cc-pVTZ//MP2/aug-cc-pVTZ//MP2/aug-cc-pVTZ//MP2/aug-cc-pVTZ//MP2/aug-cc-pVTZ//MP2/aug-cc-pVTZ//MP2/aug-cc-pVTZ//MP2/aug-cc-pVTZ//MP2/aug-cc-pVTZ//MP2/aug-cc-pVTZ//MP2/aug-cc-pVTZ//MP2/aug-cc-pVTZ//MP2/aug-cc-pVTZ//MP2/aug-cc-pVTZ//MP2/aug-cc-pVTZ//MP2/aug-cc-pVTZ//MP2/aug-cc-pVTZ//MP2/aug-cc-pVTZ//MP2/aug-cc-pVTZ//MP2/aug-cc-pVTZ//MP2/aug-cc-pVTZ//MP2/aug-cc-pVTZ//MP2/aug-cc-pVTZ//MP2/aug-cc-pVTZ//MP2/aug-cc-pVTZ//MP2/aug-cc-pVTZ//MP2/aug-cc-pVTZ//MP2/aug-cc-pVTZ//MP2/aug-cc-pVTZ//MP2/aug-cc-pVTZ//MP2/aug-cc-pVTZ//MP2/aug-cc-pVTZ//MP2/aug-cc-pVTZ//MP2/aug-cc-pVTZ//MP2/aug-cc-pVTZ//MP2/aug-cc-pVTZ//MP2/aug-cc-pVTZ//MP2/aug-cc-pVTZ//MP2/aug-cc-pVTZ//MP2/aug-cc-pVTZ//MP2/aug-cc-pVTZ//MP2/aug-cc-pVTZ//MP2/aug-cc-pVTZ//MP2/aug-cc-pVTZ//MP2/aug-cc-pVTZ//MP2/aug-cc-pVTZ//MP2/aug-cc-pVTZ//MP2/aug-cc-pVTZ//MP2/aug-cc-pVTZ//MP2/aug-cc-pVTZ//MP2/aug-cc-pVTZ//MP2/aug-cc-pVTZ//MP2/aug-cc-pVTZ//MP2/aug-cc-pVTZ//MP2/aug-cc-pVTZ//MP2/aug-cc-pVTZ//MP2/aug-cc-pVTZ//MP2/aug-cc-pVTZ//MP2/aug-cc-pVTZ//MP2/aug-cc-pVTZ//MP2/aug-cc-pVTZ//MP2/aug-cc-pVTZ//MP2/aug-cc-pVTZ//MP2/aug-cc-pVTZ//MP2/aug-cc$ pVTZ. These results indicate that the combination of the computationally cheaper M06-2X and CCSD(T) does not provide accurate results for these systems. Therefore, the Ea values calculated at the CCSD(T)/aug-cc-pVTZ//MP2/6-31+G(3df,2p) level were adopted for subsequent kinetics calculations. Notably, H-abstraction in the  $CH_2=N^{\cdot}+O_2$ ,  $CH_3CH=N^{\cdot}+O_2$ , and  $\cdot N=CHCH_2OH+O_2$  systems, as well as O<sub>2</sub> addition in the (CH<sub>3</sub>)<sub>2</sub>C=N· + O<sub>2</sub> system, remain the most favorable reaction pathways even at higher CCSD(T)/aug-cc-pVTZ//MP2/6-31+G(3df,2p) methods.

Since  $E_a$  values of the direct H-abstraction pathways are notably lower than those of the addition pathways for CH<sub>2</sub>=N·, CH<sub>3</sub>CHN· and HOCH<sub>2</sub>CHN·, the overall  $k_{O2}$  are approximated to be the rate constants of the direct H-abstraction pathways. As for the (CH<sub>3</sub>)<sub>2</sub>C=N·, the overall  $k_{O2}$  are assumed to be determined by the O<sub>2</sub>-addition pathways. The TST and master equation methods were applied to examine the kinetics of the direct H-abstraction pathway and O<sub>2</sub>-addition pathways, respectively. The calculated  $k_{O2}$  of the direct H-abstraction pathway for CH<sub>2</sub>=N· + O<sub>2</sub>, CH<sub>3</sub>CHN· + O<sub>2</sub> and HOCH<sub>2</sub>CHN· + O<sub>2</sub> reactions based on the energies calculated at the CCSD(T)/aug-cc-pVTZ//MP2/6-31+G(3df,2p) level are  $8.94 \times 10^{-18}$ ,  $1.15 \times 10^{-17}$  and  $1.38 \times 10^{-17}$  cm<sup>3</sup> molecule<sup>-1</sup> s<sup>-1</sup> at 298 K, respectively. In the case of (CH<sub>3</sub>)<sub>2</sub>C=N·, the calculated  $k_{O2}$  values of O<sub>2</sub> addition for (CH<sub>3</sub>)<sub>2</sub>C=N· is  $2.40 \times 10^{-19}$  cm<sup>3</sup> molecule<sup>-1</sup> s<sup>-1</sup>. The  $k_{O2}$  values of these four N-centered radicals formed from imines oxidation are similar to or higher than those of chain- and cyclic-like N-centered radicals from amines oxidation. Therefore, the  $k_{O2}$  values for N-centered radicals with O<sub>2</sub> varies greatly with their specific molecular structures. Previous studies have shown that the reactions of amines-derived N-centered radicals with O<sub>2</sub> can compete with their reactions with NO under typical tropospheric NO concentrations (Ma et al., 2018). Therefore, we further investigated the reactions of these four N-centered radicals with NO.

Figure 2. Schematic ZPE-corrected PES for the reactions of NO with (A)  $CH_2=N^{\bullet}$ , (B)  $CH_3CH=N^{\bullet}$ , (C)  $(CH_3)_2C=N^{\bullet}$ , and (D)  $\cdot N=CHCH_2OH$  at the CCSD(T)/6-311+G(2df,2p)//M06-2X/6-31+G(d,p) level of theory.

The calculated ZPE-corrected PES for the reactions of these four N-centered radicals with NO are presented in Figure 2. As shown in Figure 2, two types of reaction pathways are observed during the initial interaction of NO with the N-centered radicals. The first is the direct H-abstraction pathway, where NO abstracts an H atom from the = $CH_x$ - (x = 1~2) sites adjacent to the =N- and - $CH_3$  groups. These H-abstraction pathways need to overcome at least 13.35 kcal mol<sup>-1</sup>  $E_a$  values to form cyanide compounds and HNO. The second is the NO addition pathway, where NO barrierlessly addition to the =N- site of

the four N-centered radicals, forming nitrosamines adducts with different conformations depending on the approach direction of NO. Interconversion between these adducts, such as  $IM_{14-1}$  and  $IM_{14-2}$ ,  $IM_{15-1}$  and  $IM_{15-2}$ ,  $IM_{17-1}$  and  $IM_{17-2}$  proceeds with very low energy barriers of 0.53, 1.42 and 1.68 kcal mol<sup>-1</sup>, respectively.

The formed four adducts (nitrosamines) can further undergo isomerization or dissociation reactions. For  $IM_{14-1}/IM_{14-2}$ ,  $IM_{15-1}/IM_{15-2}$ ,  $IM_{16-1}$ ,  $IM_{17-1}/IM_{17-2}$ , three, five, three and four H-shift pathways are identified, respectively. It is observed that H-shifts from the  $=CH_x-(x=1\sim2)$  sites adjacent to the =N- and  $-CH_3$  groups to the O-atom of the -NNO group are the most favorable. However, the formed four adducts need to overcome high barriers to isomerize or dissociate into fragmentation products. This mechanism is analogous to amines-derived N-centered radicals with NO. In addition, the main reaction pathway remains even at high computational methods (see Table S5 and S6).

To maintain consistency with the N-centered +  $O_2$  reactions, the reaction energies calculated at the CCSD(T)/aug-cc-pVTZ/MP2/6-31+G(3df,2p) level are used to calculate the reaction rate constant ( $k_{NO}$ ) for the reactions of four N-centered radicals with NO. The calculated reaction rate constants for NO addition pathways for the reactions of  $CH_2=N^{\cdot}+NO$ ,  $CH_3CHN^{\cdot}+NO$ ,  $(CH_3)_2C=N^{\cdot}+NO$  and  $HOCH_2CHN^{\cdot}+NO$  are  $2.20\times10^{-16}$ ,  $1.42\times10^{-12}$ ,  $1.09\times10^{-10}$  and  $4.30\times10^{-11}$  cm<sup>3</sup> molecule<sup>-1</sup> s<sup>-1</sup> at 298 K, respectively. These are much higher than those for the corresponding direct H-abstraction pathways  $(1.85\times10^{-21}, 2.87\times10^{-21}, 5.57\times10^{-27}$  and  $1.04\times10^{-19}$  cm<sup>3</sup> molecule<sup>-1</sup> s<sup>-1</sup>). Therefore, the  $k_{NO}$  can be assumed to be equal to the reaction rate constant for the addition pathways. The calculated  $\Gamma$  values for the nitrosamine (IM<sub>14-1</sub>, IM<sub>15-1</sub>, IM<sub>16-1</sub> and IM<sub>17-1</sub>) for the reactions of  $CH_2=N^{\cdot}+NO$ ,  $CH_3CH=N^{\cdot}+NO$ ,  $CH_3C=N^{\cdot}+NO$  and  $N=CHCH_2OH+NO$  are 0.08%, 6.22%, 52.56% and 35.32% at 1 atm and 298 K, respectively. Consequently, except for  $CH_2=N^{\cdot}+NO$  and  $CH_3CH=N^{\cdot}+NO$ , the reactions of  $CH_3C=N^{\cdot}+NO$  and  $CH_3CH=N^{\cdot}+NO$  and  $CH_3CH=N^{\cdot}+NO$  and  $CH_3CH=N^{\cdot}+NO$  and  $CH_3CH=N^{\cdot}+NO$  and  $CH_3CH=N^{\cdot}+NO$  and  $CH_3CH=N^{\cdot}+NO$  are significantly lower than those of PZ-N (99.97%), Monoethanolamine (MEA)-N (86%) and  $CH_3NH^{\cdot}+(-60\%)$  with NO.

Using the calculated  $k_{O2}$  and  $k_{NO}$ , we assessed the competition between  $O_2$  and NO for the four N-centered radicals. The required concentrations of NO ([NO]) to equalize the pseudo-first-order rate constants for N-centered radicals with  $O_2$  are  $8.1 \times 10^6$ ,  $1.7 \times 10^3$ , 0.44, and 64 ppb, respectively. For  $CH_2=N^{\cdot}$  and  $CH_3CH=N^{\cdot}$ , the required [NO] are very high, far exceeding the typical [NO] encountered in the atmosphere. Therefore,  $CH_2=N^{\cdot}$  and  $CH_3CHN^{\cdot}$  are expected to primarily react with  $O_2$  to form HC=N and  $CH_3C=N$ , respectively, consistent with previous studies of  $CH_2=NH$ . In contrast, for  $(CH_3)_2C=N^{\cdot}$  and  $N=CHCH_2OH$ , the required [NO] are achievable under polluted atmospheric conditions, suggesting that both  $(CH_3)_2C=N^{\cdot}$  and  $N=CHCH_2OH$  primarily react with NO to form nitrosamines. To the best of knowledge, this study represents the first report demonstrating that N-centered radicals formed from imines oxidation can lead to carcinogenic nitrosamine. However, the yields of nitrosamine formation is highly dependent on the specific structures of N-centered radicals.

### 4 Implications

Similar to previous studies on amines + ·Cl reaction systems, the reaction of imines containing =NH

(CH<sub>2</sub>=NH, CH<sub>3</sub>CH=NH, (CH<sub>3</sub>)<sub>2</sub>C=NH, and HN=CHCH<sub>2</sub>OH) with ·Cl predominantly yield N-centered radicals. The calculated  $k_{Cl}$  values for the reactions of CH<sub>2</sub>=NH, CH<sub>3</sub>CH=NH, CH<sub>3</sub>N=CH<sub>2</sub>, (CH<sub>3</sub>)<sub>2</sub>C=NH, and HN=CHCH<sub>2</sub>OH are  $4.50 \times 10^{-11}$ ,  $2.72 \times 10^{-10}$ ,  $7.32 \times 10^{-11}$ ,  $4.48 \times 10^{-10}$  and  $1.26 \times 10^{-10}$  cm<sup>3</sup> molecule<sup>-1</sup> s<sup>-1</sup> at 298 K and 1 atm, respectively. Among the five imines studied, only the reaction kinetics of CH<sub>2</sub>=NH and CH<sub>3</sub>N=CH<sub>2</sub> with ·OH are available (CH<sub>2</sub>=NH: 3.00 × 10<sup>-12</sup> cm<sup>3</sup> molecule<sup>-1</sup> s<sup>-1</sup>, 320  $\text{CH}_3\text{N}=\text{CH}_2$ :  $3.70 \times 10^{-12} \text{ cm}^3 \text{ molecule}^{-1} \text{ s}^{-1}$ ). With this data, we can estimate the contribution of  $\cdot \text{Cl}$  to the degradation of CH<sub>2</sub>=NH and CH<sub>3</sub>N=CH<sub>2</sub> based on the ·Cl concentration ([·Cl]). In the marine boundary layer, the [·Cl] is typically 1-10% of the [·OH]. The contribution of ·Cl relative to ·OH  $(k_{CI}[\cdot CI]/k_{OH}[\cdot OH])$  to the transformation of  $CH_2$ =NH and  $CH_3$ N= $CH_2$  are estimated to be 15% - 150% and 20% - 200%, respectively. Furthermore, the contribution of ·Cl relative to ·OH to the formation of 325  $CH_2=N$  is estimated at 34% - 340% (estimated by  $k_{CI}[\cdot CI] \times \Gamma_{N,CI} / k_{OH}[\cdot OH] \times \Gamma_{N,OH}$ , where  $\Gamma_{N,CI}$  and  $\Gamma_{N,OH}$  represent the yields of N-centered radicals from the reactions initiated by ·Cl and ·OH, respectively) This clearly demonstrates the significant role of ·Cl in the transformation of CH<sub>2</sub>=NH and CH<sub>3</sub>N=CH<sub>2</sub>. Although complete data on the reactions of other imines (CH<sub>3</sub>CH=NH, (CH<sub>3</sub>)<sub>2</sub>C=NH, and HN=CHCH2OH) with ·OH are lacking, it is reasonable to believe that ·Cl also plays a crucial role in 330 their transformation based on their high reaction rate constants.

Unlike N-centered radicals generated from amines oxidation, this study reveals that both  $k_{O2}$  and  $k_{NO}$  values for N-centered radicals generated from imines oxidation are strongly dependent on their specific structures, ultimately affecting nitrosamine formation. This study provides the first evidence that N-centered radical formed from imines oxidation can yield nitrosamines under polluted atmospheric conditions. Therefore, to comprehensively evaluate the formation of nitrosamines from imines, further investigations into the reactions of imine-derived N-centered radicals with  $O_2$  and NO are warranted.

*Data availability.* All data were available in the main text or supplementary materials. The other relevant data are available upon request from the corresponding authors.

Author contribution. HBX and MFF designed research; XQ, MFF and HBX performed research; XQ, MFF, LC, ZQJ and HBX analyzed data; XQ and MFF wrote the paper; XQ, MFF, CJW and HBX reviewed and revised the paper.

Competing interests. The authors declare that they have no conflict of interest.

Acknowledgements. The study was supported by the National Natural Science Foundation of China 345 (22206020, 22236004, 22176022), the National Key Research and Development Program of China (2022YFC3701000), and the Sugon Supercomputing Center.

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
