# Peer review of "Chlorine Radical-Initiated Atmospheric Oxidation of Imines: Implications for Structural Influence on the Nitrosamine Formation"

_EGUsphere, 2025_

## Author Comment (AC1)

**General comment**

*Xu et al. systematically studied the reaction mechanisms and kinetics of the ·Cl + imines reactions using computational methods. The authors found that imine-derived N-centered radicals exhibits various fates under tropospheric conditions, which significantly differs from the behavior of previously reported amines. This is an interesting study and the paper is a nice addition to the literature on the oxidation of organic nitrogen compounds. I believe this paper is fitting well in ACP. I can recommend publication after the following comments have been addressed.*

**Response:** We highly appreciate your insightful and helpful comments on our manuscript. We have revised the manuscript to enhance its quality.

**Special Suggestions and Comments**

(1) *I'm confused why two different kinetic simulation software programs are used for the initial and subsequent reactions*

**Response:** Thanks for the comment. In fact, all the reaction rate constants were calculated using MultiWell 2014.1 software. In the revised manuscript, the sentences have change to:

MultiWell 2014.1 software was employed to simulate the reaction kinetics (Barker et al., 2014; Barker, 2001).

(Please see Lines 125-126)

(2) *Not clear to this reviewer which reaction steps were simulated using MultiWell program, and which were simulated using the MESMER program.*

**Response:** We appreciated the comment and agree that the statement is not clear here. As discussed in the response on the comment (1), we have rewritten the sentence. Please see Lines 125-126.

(3) *I am missing labels (A)~(E) in the Table 1.*

**Response:** Thanks. We have corrected it. (Please see Table 1)

(4) *The branch ratio in Table 1 does not appear to be 1.*

**Response:** Thanks. We have corrected it. (Please see Table 1)

(5) *The overall reaction rate constants in the abstract are reported with a unit factor of $\times 10^{-11}$ cm$^3$ molecule$^{-1}$ s$^{-1}$, while the same constants in Table 1 use a factor of $\times 10^{-10}$ cm$^3$ molecule$^{-1}$ s$^{-1}$. Please ensure consistency throughout the manuscript.*

**Response:** Thanks. We have corrected it. (Please see Table 1)

(6) *Missing reference for Molclus program.*

**Response:** Thanks. We have corrected it. (Please see Line 98 and reference Lu, 2025)

(7) *The parameters used in the long-range transition-state theory were not shown in the manuscript.*

**Response:** As discussed in the response on the Reviewer 1 comment (5), we added the parameters used in the LRTST to Table S1 in the revised SI.

(8) *I suggest some important results can be moved from SI to the main text, for example, Figure S3 and S4.*

**Response:** Thanks for the comment and suggestion! We have combined Figures S3 and S4 into Figure 1 and included it in the revised manuscript.

[Figure]

**Figure 1.** Reaction rate constants ($k_{Cl}$) for the reactions of five imines (a) and branching ratios ($\Gamma$) for the species involved in the reactions of (b) $CH_2=NH$, (c) $CH_3CH=NH$, (d)

CH$_3$N=CH$_2$, (e) (CH$_3$)$_2$C=NH, and (f) HN=CHCH$_2$OH initiated by ·Cl in the temperature range of 260 - 300 K and 1 atm.

---

## Author Comment (AC2)

The authors would like to thank the reviewers for this discussion and their constructive comments, corrections and suggestions that ensued. We have carefully replied to all their comments and have made improvements to the paper based on their suggestions. Our replies to their comments are in blue.

**Reviewer 1**

**General comment**

*This manuscript provides a comprehensive theoretical investigation into the ·Cl-initiated reaction mechanisms of five representative imines. By integrating quantum chemical calculations and kinetic calculations, the study elucidates the reaction pathways, rate constants, and the fate of generated radicals, with particular emphasis on the formation of N-centered radicals and their potential role in nitrosamine formation. This work fills a gap in our understanding of ·Cl-initiated atmospheric oxidation of imines and has important implications for atmospheric chemistry. Overall, the manuscript is well written and clearly presented. In my opinion, it can be accepted for publication after the authors address the following issues*

**Response:** We highly appreciate your insightful and helpful comments on our manuscript. We have revised the manuscript to enhance its quality.

**Special Suggestions and Comments**

(1) *The authors cited many references to introduce the significance sources of ·Cl. I think some significant works also need to read and properly cited in the revised manuscript. (For example, Environ. Sci. Technol.2025, 59, 12775; Environ. Sci. Technol. 2025, 59, 9167; Environ. Sci. Technol. 2025, 59, 6155)*

**Response:** Thanks for the suggestion. The suggestions have been adopted. We have cited the references mentioned above in the revised manuscript. (Please see Lines 33-38 and references Ma et al., 2025; Cooke et al., 2025; Chen et al., 2025)

(2) *Missing references for the sentence 'Daytime ·Cl concentrations have been reported to reach up to $10^6$ molecule $cm^{-3}$'.*

**Response:** Thanks. We have corrected it. (Please see Lines 40-41 and references Young et al., 2014; Li et al., 2025; Wang et al., 2023)

(3) *Computational Details: Missing reference for Molclus program.*

**Response:** Thanks. We have corrected it. (Please see Line 98 and reference Lu, 2025)

(4) *They should make a bit more clear (in section 2.2 and 2.3) why two different methods (MP2 and M06-2X) and programs (MultiWell and MESMER) were used.*

**Response:** We appreciate your suggestion and agree that the statement is not clear here. In fact, all reaction rate constants were calculated using MultiWell 2014.1 software, and the description of Mesmer has been removed. In the revised manuscript, the sentences have change to:

For the ·Cl-initiated reactions of five imines, geometry optimizations and harmonic

vibrational frequency calculations were performed at the MP2/6-31+G(3df,2p) level, followed by single-point energy evaluations at the CCSD(T)/aug-cc-pVTZ level. This is consistent with our previous work on ·Cl-initiated reactions of ONCs systems, where the combination of MP2 and CCSD(T) methods has proven to yield reliable energies. Considering the substantially increased computational cost for the subsequent reactions of the resulting imine radicals, we employed the M06-2X/6-31+G(d,p) level for geometry optimizations and frequency calculations, followed by CCSD(T)/6-311+G(2df,2p) single-point energy evaluations. The M06-2X functional combined with CCSD(T) method has successfully been applied to predict radical + $O_2$/NO reactions.

MultiWell 2014.1 software was employed to simulate the reaction kinetics (Barker et al., 2014; Barker, 2001).

(Please see Lines 105-117; 125-126)

(5) *Computational Details: please give the parameters used in the long-range transition-state theory with a dispersion force potential.*
**Response:** We appreciated the suggestion. We added the parameters used in the LRTST to Table S1 in the Lines 135-137 and revised SI.

**Table S1.** Polarizabilities ($\alpha$) and the first ionization potentials ($I$) used in the long-range transition state theory (LRTST).

| Species | $\alpha$ ($a_0^3$) | $I$ (eV) |
|---|---|---|
| $CH_2$=NH | 21.21* | 9.52* |
| ·CH=NH ($P_{1-1}$) | 20.86* | 6.64* |
| ·CH=NH ($P_{1-2}$) | 22.04* | 6.83* |
| $CH_2$=N· ($P_{1-3}$) | 19.30* | 7.17* |
| $CH_3$CH=NH | 33.69* | 9.42* |
| ·$CH_2$CH=NH ($P_{2-1}$) | 29.58* | 7.46* |
| ·$CH_2$CH=NH ($P_{2-2/3}$) | 35.90* | 9.01* |
| ·$CH_3$C=NH ($P_{2-4}$) | 33.75* | 6.11* |
| $CH_3$CH=N· ($P_{2-5}$) | 31.05* | 6.82* |
| $CH_3$N=$CH_2$ | 34.01* | 9.07* |
| $CH_3$N=CH· ($P_{3-1}$) | 35.32* | 5.89* |
| $CH_3$N=CH· ($P_{3-2}$) | 36.69* | 6.02* |
| ·$CH_2$N=$CH_2$ ($P_{3-3}$) | 34.04* | 6.28* |
| ·$CH_2$N=$CH_2$ ($P_{3-4/5}$) | 36.60* | 6.71* |
| $(CH_3)_2$C=NH | 45.41* | 9.07* |

| | | |
|---|---|---|
| ·(CH$_3$)(CH$_2$)=NH (P$_{4-1}$) | 41.16* | 6.95* |
| ·(CH$_3$)(CH$_2$)=NH (P$_{4-2/3}$) | 47.00* | 8.69* |
| ·(CH$_3$)(CH$_2$)=NH (P$_{4-4}$) | 41.17* | 7.01* |
| ·(CH$_3$)(CH$_2$)=NH (P$_{4-5/6}$) | 47.09* | 8.91* |
| (CH$_3$)(CH$_2$)=N· (P$_{4-7}$) | 42.32* | 6.13* |
| HN=CHCH$_2$OH | 37.71* | 8.99* |
| ·HN=CHCHOH (P$_{5-1/2}$) | 41.70* | 7.51* |
| ·HN=CCH$_2$OH (P$_{5-3}$) | 37.61* | 6.48* |
| ·N=CHCH$_2$OH (P$_{5-4}$) | 35.25* | 6.78* |
| HN=CHCHO· (P$_{5-5}$) | 34.61* | 7.40* |
| ·Cl | 14.71[#] | 12.97[#] |
| HCl | 16.97[#] | 12.74[#] |

*$\alpha$ and I were calculated at CCSD(T)/aug-cc-pVTZ//MP2/6-31+G(3df,2p) level of theory, respectively
[#]Obtained from the NIST database

(6)  *'CCSD(T)/6-311+g(2df,2p)' should be 'CCSD(T)/6-311+G(2df,2p)'.*
**Response:** According to the reviewer's suggestions, the corresponding corrections were done. (Please see Lines 163-165)

(7)  *Please mention the pressure in the figure caption.*
**Response:** We have added the pressure in the Figure captions. (Please see Figure 1)

(8) *The branching ratio in Table 1 does not appear to be 1. Please check carefully.*
**Response:** Thanks. We have corrected it. (Please see Table 1)

(9)  *Page 6, line 170: Missing Figure in SI.*
**Response:** Yes. We have added the structures of TS$_{5-4}$ and TS$_{5-5}$ to Figure S3 in the revised SI.

[Figure]

**Figure S3.** Optimized geometries of TSs for TS$_{5-4}$ (a) and TS$_{5-5}$ (b) in the ·Cl + HN=CHCH$_2$OH reactions. The dashed red line in (a) indicates the intramolecular hydrogen bond. The distances are in Å

(10) *Page7, line 220: 'intramolecular' should be 'unimolecular'.*
**Response:** We have corrected it. (Please see Lines 232-233)

(11) *Page 10, line 305-309, Missing references.*
**Response:** Thanks. We have corrected it. (Lines 320-323 and references Ren, 2003; Bunkan et al., 2014.)